# Long-Term Kinetics of Serum Galactomannan during Treatment of Complicated Invasive Pulmonary Aspergillosis

**DOI:** 10.3390/jof9020157

**Published:** 2023-01-24

**Authors:** Athanasios Tragiannidis, Christina Linke, Carlos L. Correa-Martinez, Heidrun Herbrüggen, Frieder Schaumburg, Andreas H. Groll

**Affiliations:** 1Infectious Disease Research Program, Center for Bone Marrow Transplantation and Department of Pediatric Hematology/Oncology, Children’s University Hospital Münster, 48149 Münster, Germany; 22nd Department of Pediatrics, Aristotle University of Thessaloniki, AHEPA Hospital, 54636 Thessaloniki, Greece; 3Department of Medical Microbiology, University Hospital Münster, 48149 Münster, Germany

**Keywords:** aspergillosis, children, cancer, transplantation, leukemia, treatment, antigen, galactomannan, kinetics, monitoring

## Abstract

Several studies have evaluated the serum galactomannan (GM) antigen assay in pediatric patients, and there is convincing evidence for its usefulness as a diagnostic tool for invasive *Aspergillus* infections in patients with acute leukemias or post allogeneic hematopoietic cell transplantation (HCT). Less is known about the utility of the assay in monitoring responses to treatment in patients with established invasive aspergillosis (IA). Here, we present the long-term kinetics of serum galactomannan in two severely immunocompromised adolescents with invasive pulmonary aspergillosis (IPA) who were cured after complicated clinical courses. We also review the utility of the GM antigen assay in serum as a prognostic tool around the time of diagnosis of IA and as a biomarker to monitor disease activity in patients with established IA and assess responses to systemic antifungal therapy.

## 1. Introduction

Galactomannan (GM) is a polysaccharide that makes up a major part of the cell wall of most *Aspergillus* spp. and *Penicillium* spp. and in lower amounts of some other fungi, including *Fusarium* spp., *Scedosporium* spp., *Alternaria* spp., and *Histoplasma* spp. It is released during invasive hyphal growth and can be detected in serum and bronchoalveolar lavage (BAL) by an enzyme immunoassay approved by the Food and Drug Administration (FDA) [1,2]. GM is detected by enzyme-linked immunosorbent assay (ELISA) performed with monoclonal anti-GM antibodies via optical density (OD) values. GM levels, expressed as the GM index (GMI), are assessed using a semi-quantitative approach by calculating the ratio of a clinical sample’s OD to that of 1 ng of a GM standard in order to calculate the optical density index (ODI). The ODI is interpreted as a negative (<0.5) or positive (≥0.5) result [3].

In neutropenic patients at high risk for developing invasive aspergillosis (IA), the diagnostic specificity and sensitivity of a positive GM test in serum are above 80% in both adults and children [4]. A positive GM test in serum is included as a microbiological criterion of probable IA in the current European Organization for Research and Treatment of Cancer and among the definitions of invasive fungal diseases by the Mycoses Study Group Education and Research Consortium (EORTC/MSGERG) [5]. Testing GM in serum is included in the current guidelines for the diagnosis and management of IA in children and adults with hematological malignancies or following allogeneic haematopoietic cell transplantation (HCT) [6,7,8,9].

While a number of reports have described the use of the GM antigen assay in serum as a prognostic tool around the time of diagnosis, considerably less is known about its utility as a biomarker to assess responses to systemic antifungal therapy or to monitor disease activity in patients with established IA [10,11]. Here, we report the long-term kinetics of serum GM in two severely immunocompromised adolescents with invasive pulmonary aspergillosis (IPA) who were cured after complicated clinical courses and prolonged antifungal treatment.

## 2. Case Reports

### 2.1. Patient 1

Patient 1 was a 15-year-old girl with refractory acute myeloid leukemia (AML) and limited response to second-line therapies, including clofarabine, gemtuzumab, and sorafenib. When first radiological signs of invasive pulmonary mold infection occurred on high-resolution computed tomography (HR-CT), she had a 3-month history of granulocytopenia and presented with persistent fever despite the use of broad-spectrum antibiotics. Antifungal prophylaxis with voriconazole (300 mg IV BID) was augmented with the addition of liposomal amphotericin B (LAMB) (3 mg/kg QD) and caspofungin (50 mg QD). Repeat CT-scans revealed growth of the two existing lesions with a rising halo sign. Imaging of the central nervous system (CNS) at diagnosis of the invasive mycoses and throughout was within normal limits. Consecutive GM antigen serum indices were between 8.4 and 11.9. Bronchoscopy with bronchoalveolar lavage (BAL) was not performed since the diagnosis of probable IPA was established based on clinical, mycological (GM), and radiological (CT-scan) criteria and the patient was in unstable condition. Due to persistent granulocytopenia, the patient received granulocyte-colony stimulating factor (G-CSF) and granulocyte transfusions on two occasions with concomitant increase in pulmonary opacification but no change in GM antigen serum index.

Six weeks after diagnosis of IPA, matched related allogeneic HCT was performed in aplasia following conditioning with treosulfan and melphalan and standard post-transplant immunosuppression with methotrexate and cyclosporine A. Nine days post peripheral blood stem cell infusion, stable hematopoietic engraftment (absolute neutrophil count; ANC > 500/µL) was documented, followed by a rapid decrease in the GM antigen serum index to 1.0 at day +19. CT imaging around that time showed a morphological change with the occurrence of an air crescent sign. Along with a steady clinical stabilization, antifungal therapy was stepwise reduced to voriconazole (300 mg PO BID).

Five weeks post transplantation, a transient flare of the GM antigen serum index of up to 9.0 was observed in the wake of acute graft-vs.-host disease (GVHD) of the skin, corticosteroid treatment, and cytomegalovirus (CMV) reactivation. Clinically, the patient presented with new fever and transient respiratory distress, modest pleural effusion, but no new inflammatory infiltrates. Granulocyte counts were stable and varied between 2000 and 4000/μL. Three months after transplantation, the patient developed gastrointestinal GVHD with a prolonged clinical course, requiring augmented immunosuppression with corticosteroids in changing doses (prednisolone or methylprednisolone, 1–2 mg/kg/day) for a period of three months. No impact on GM antigen serum indices was noted, and signs and symptoms of IPA resolved under continuous treatment with voriconazole with the disappearance of active lesions and a negative GM antigen index in serum five months post-transplant. Changes in serum GM antigen index over time and in correlation with neutrophil and CD4 + T-lymphocyte counts are shown in Figure 1A, and exemplary CT imaging results are shown in Figure 1B.

### 2.2. Patient 2

Patient 2 was a 13-year-old girl with childhood myelodysplastic syndrome/refractory cytopenia (MDS/RCC) and matched, unrelated, allogeneic HCT with bone marrow following conditioning with fludarabine, thiotepa, and anti-thymocyte globuline (ATG) and standard post-transplant immunosuppression with methotrexate and cyclosporine A. Probable IPA was diagnosed during aplasia at day +25 post-transplant by HR-CT, detection of GM in serum (index range, 2.6–4.3) and BAL fluid (7.1), and molecular detection of *Aspergillus fumigatus* in BAL fluid by polymerase chain reaction (PCR). Imaging of the CNS was normal at diagnosis and throughout. Empirical therapy with liposomal amphotericin B (3 mg/kg QD) was augmented with voriconazole (200 mg IV BID) and caspofungin (50 mg QD). After three separate granulocyte transfusions, hematopoietic engraftment was documented at day +43. Antifungal therapy was reduced to voriconazole. Repeat CT-scans showed decreasing volume of all existing lesions and sequestrations and the GM antigen serum indices fell below 1.0.

The patient’s further course was complicated by acute GVHD of the skin and systemic CMV and adenovirus (ADV) reactivation, requiring prolonged and augmented immunosupression with corticosteroids and use of antiviral agents, respectively. During that time, new pulmonary infiltrates occurred on HR-CTs and antifungal therapy was escalated with the addition of caspofungin (50 mg QD). Due to secondary graft failure, the patient received a stem cell boost without conditioning, resulting in persisting hematopoietic engraftment within 10 days. Around that time, GM antigen serum indices showed a transient increase.

Subsequently, GVHD and virus reactivations required augmented immunosuppressive treatment with glucocorticosteroids and myelosuppressive antiviral therapy for several months. Permanent stabilization of granulocyte counts was observed 3 months after the stem cell boost and normal CD4 + T-lymphocyte counts after eight months. During that period, the GM serum index showed high variability without clear correlation with pulmonary CT findings. Nine months after the stem cell boost, GM antigen serum indices dropped permanently to under 1.0. Imaging by HR-CT showed successive reduction in infiltrative and cavitary pulmonary lesions. All lesions eventually disappeared and the patient is in continuous remission and doing well at +3 years post-transplant. Changes in the serum GM antigen index in comparison with treatment phases over time and in correlation with neutrophil and CD4 + T-lymphocyte counts and CD4 T-cell values are shown in Figure 2A, and exemplary CT imaging results are shown in Figure 2B.

## 3. Discussion

Invasive aspergillosis (IA) continues to be an important cause of morbidity in children and adolescents with allogeneic HCT, requiring considerable resources for prevention, diagnosis, and management, and is associated with significant reductions in overall survival [12,13,14]. While definitive microbiological diagnosis of IA is a major challenge, GM antigen in serum has become a universally accepted diagnostic biomarker in high-risk symptomatic adult and pediatric patients in combination with clinical and radiographic findings [4,5,15]. Although several studies have investigated the use of the GM antigen assay in serum as a diagnostic tool in pediatric patients [4,16], limited data exist on its usefulness for assessing therapeutic responses and guiding therapeutic decisions in this population. However, a reliable dynamic tool to assess treatment responses would greatly improve the care of affected patients and expedite clinical trials since responses to treatment of IA and specifically, IPA, occur over extended periods of time and clinical and radiographic findings are often unspecific and variable [17,18,19].

GM is present in the cell wall of *Aspergillus* spp. hyphae and its detection in serum is an index of hyphal tissue invasion and strongly indicative of invasive *Aspergillus* disease. In a well-established and validated experimental model of IPA in persistently neutropenic rabbits, the release of GM was correlated with progression of disease and its resolution with effective antifungal therapy [20,21,22,23,24,25,26]. In this model, GM could be detected in serum by ELISA as early as 12 h post-inoculation, corresponding with the demonstration of hyphal tissue invasion by histopathology. Without treatment, serum GM concentrations continue to increase and eventually reach a plateau [22]. With effective antifungal treatment, GM concentrations in serum decline in parallel with lung weight, pulmonary infarction score, and residual fungal tissue burden, with similar utility in the evaluation of dose- and time dependent effects at end of treatment as these conventional outcome measures [20,21,22,23]. Pharmacokinetic/pharmacodynamic assessments of the kinetics of GM in serum in the neutropenic rabbit model, which have estimated exposure-response relationships and bridged the findings to humans, have demonstrated the potential utility of this readily available biomarker for dynamic monitoring and estimating responses to treatment in neutropenic patients with IA [23,25,26]. Of note, in the equivalent model in non-neutropenic, cyclosporine-methylprednisolone immunosuppressed rabbits, GM concentrations in serum were consistently lower, corresponding to shorter and fewer hyphae in lung tissue and an intensive neutrophil response to *Aspergillus* [24]. In both models, GM concentrations in BAL were typically high throughout the course of infection [24], reflecting hyphal growth in the airways and not necessarily hyphal tissue invasion [3,24]. These additional experimental findings indicate a limited usefulness of the GM assay in BAL for estimating responses and a reduced utility of the GM assay in serum for non-neutropenic immunosuppressed subjects. The latter is of particular importance since neutropenic HCT recipients with IA eventually transition to the non-neutropenic state and a relevant proportion of IA occurs in the non-neutropenic, immunosuppressed state [12,13,14].

Several clinical studies have shown the principal correlation of GM in serum with endpoints outcomes of proven or probable IA in neutropenic patients with hematological malignancies or allogeneic HCT [10,27,28,29,30], and for predicting response and survival using the kinetics of serum GM within the first one to two weeks of treatment [3,11,31,32,33,34,35,36,37]. For example, in 158 adult patients with positive baseline serum GM and serial GM assessments enrolled in a phase 3 clinical trial for the treatment of invasive mold disease [38], increases in the GM index by day seven of >0.25 units from baseline increased the risk of death compared to those with no increase or increases <0.25 (hazard ratio, 9.766; 95% confidence interval [CI], 4.3–21.9; *p* < 0.0001). For each unit decrease by day 7 from baseline, the odds of successful therapy doubled (odds ratio, 2.1; 95% CI, 1.1–3.9) [3]. Similarly, in 71 patients enrolled in the multicenter, phase 3 Global Aspergillosis Study with a baseline serum GM index of ≥0.5, the week one GM index was significantly lower for eventual responders (0.62 ± 0.12 vs. 1.15 ± 0.22 in non-responders; *p* = 0.035) [39]. A GM index reduction of >35% between baseline and week one predicted a probability of satisfactory clinical response. Among the 131 IA patients with a pretreatment serum GM index of <0.5, a rising absolute GM index to >0.5 at week two, despite antifungal treatment, heralded a poor clinical outcome [34].

Changes in the serum GM index from baseline have also been correlated with MSG/EORTC outcome criteria at 6 and 12 weeks, mortality at 12 weeks, and autopsy findings. The GM index-based outcome proposed from these correlations was defined as GMI negativity for at least two weeks after the first positive value, without new pulmonary and extrapulmonary lesions, and lack of findings of IA on autopsy [27,28,29,40,41]. Current practice guidelines of the Infectious Diseases Society of America (IDSA), as well as the joint clinical guidelines of the European Society for Clinical Microbiology and Infectious Diseases, the European Confederation of Medical Mycology, and the European Respiratory Society (ESCMID/ ECMM/ ERS), recommend serial monitoring of serum GM in patients with hematological malignancies and allogeneic HCT to monitor disease progression, therapeutic response, and predict the outcome of IA [6,7]. An expert panel recently convened to develop criteria for the management of IA primarily for use in designing clinical trials with new antifungal agents [19]. According to these criteria, they defined primary treatment failure at, or after, 8 days of primary antifungal treatment as: increasing serum GM index or GM positivity in serum when the antigen was previously undetectable, sudden clinical deterioration, or detection of a new, clearly distinct site of infection. Furthermore, treatment failure was defined at, or after, 15 days of primary antifungal treatment as: patient is clinically stable but shows persistently elevated serum GM index ≥2 or increasing compared with baseline, or if the original lesions on CT or other imaging show progression by >25% in size with no apparent change in immune status [19].

While there is no biological rationale to suggest differences in the pediatric population, few studies on GM in serum for predicting responses to treatment or outcome of IA have been reported in children and adolescents with hematological malignancies and/or allogenic HCT. In a case control study involving 191 children with high-risk febrile neutropenia of whom 107 received antifungals, a difference threshold in the GM index of ≥0.3 was significantly associated with antifungal treatment modification (OR 5.0, 95% CI 1.0–25.7, *p* = 0.04) [42]. In a retrospective study that included 45 proven/probable cases of IA of whom 15 eventually died, serum GM values were higher in the fatality group than in the survival group during the entire period of antifungal therapy, and serum GM values at one week after the start of antifungal therapy were most significantly associated with mortality. A serum GM index  >1.50 at one week post-diagnosis had a sensitivity and specificity of 61.5% and 89.3%, respectively, in predicting mortality within 12 weeks after the start of antifungal therapy [43]. Similar to a larger study in adult patients receiving isavuconazole [17], pharmacokinetic/pharmacodynamic (PK/PD) analyses to study the drug exposure necessary for negativity of serum GM at end of therapy have been conducted in a few pediatric patients with IA receiving treatment with voriconazole. A previous study reported (AUC/EC50)/15.4 predicted terminal galactomannan (*p* = 0.003) and an AUC/EC_50_ ratio of >6 suggested a lower terminal galactomannan level (*p* = 0.07), demonstrating the principal ability of such linked PK-PD models for individualized drug dosing to achieve optimal suppression of galactomannan levels [44].

Whereas the follow-up time in clinical studies is limited to a maximum of approximately 40 days [3,25], the cases presented in this report provide a unique opportunity to visualize the dynamics of GM kinetics in individual patients with protracted courses of IA during common complications post-allogeneic HCT over a time period of 6 to 18 months from diagnosis to end of treatment.

In patient 1 (refractory AML and probable IPA diagnosed six weeks prior to allogeneic HCT under monitored prophylaxis with voriconazole; Figure 1), GM levels in serum stayed elevated for more than six weeks despite triple combination therapy in parallel with an increasing lesion volume on CT imaging. GM indices decreased with neutrophil engraftment along with sequestration of the lesion and occurrence of the air crescent sign. GM indices again increased after de-escalation to voriconazole despite stable neutrophil engraftment but in parallel with new respiratory symptoms without correlating CT findings during an episode of acute GVHD, steroid treatment, and CMV-reactivation. GM indices decreased following successful management of GVHD and control of CMV reactivation under continued voriconazole treatment, remained low with stable imaging findings during a later, prolonged episode of GVHD and protracted steroid treatment, and eventually became negative with discontinuation of immunosuppression and immuno-reconstitution.

In patient 2 (MDS and probable IPA diagnosed three weeks after allogeneic HCT under empirical therapy with liposomal amphotericin B, Figure 2), GM levels in serum decreased under triple combination therapy following engraftment in parallel with a decreasing volume and sequestration of lung lesions on CT imaging. Following de-escalation to voriconazole, GM indices increased along with new pulmonary lesions following the occurrence of GVHD, steroid treatment, CMV and ADV reactivation, and neutropenia due to secondary graft failure. GM indices decreased after a stem cell boost with increasing neutrophil counts under a combination of voriconazole and caspofungin. They increased again without clear clinical or radiographic correlations during another episode of neutropenia under ongoing steroids and antiviral treatment to eventually become negative with discontinuation of immunosuppression and immune reconstitution.

Overall, the analysis of the two cases strongly suggests a correlation between the GM kinetics in serum and the kinetics of the neutrophil counts and disease control, as reflected by pulmonary imaging findings and a more subtle association with GVHD, ensuing steroid treatment, and viral reactivations. This reflects the experimental findings of the strong association of GM shedding in serum with invasive hyphal growth in the context of neutropenia [22,24] and the predominant impact of neutrophil recovery on the prognosis of IA acquired during profound and prolonged granulocytopenia [12]. However, apart from the presence and function of neutrophils, the kinetic profile of GM in serum is complex and may be affected by other factors, including the involved *Aspergillus* species, fungal burden, antifungal drug exposure, the minimum inhibitory concentration (MIC) of the invading pathogen, and the patient’s renal and hepatic function, respectively [3,45,46].

Taken together, the presented cases support the principal utility of monitoring serum GM for assessing responses to treatment and disease activity in protracted courses of IPA before and after allogeneic HCT. In conjunction with clinical and imaging findings, the kinetics of serum GM may be an important adjunct in guiding treatment and serial monitoring is strongly suggested in affected patients. In the absence of validated breakpoints for predicting treatment failure, the algorithms proposed in relevant clinical guidelines [6,7] and position papers may be considered in decision-making for individual patients [19].

## Figures and Tables

**Figure 1 jof-09-00157-f001:**
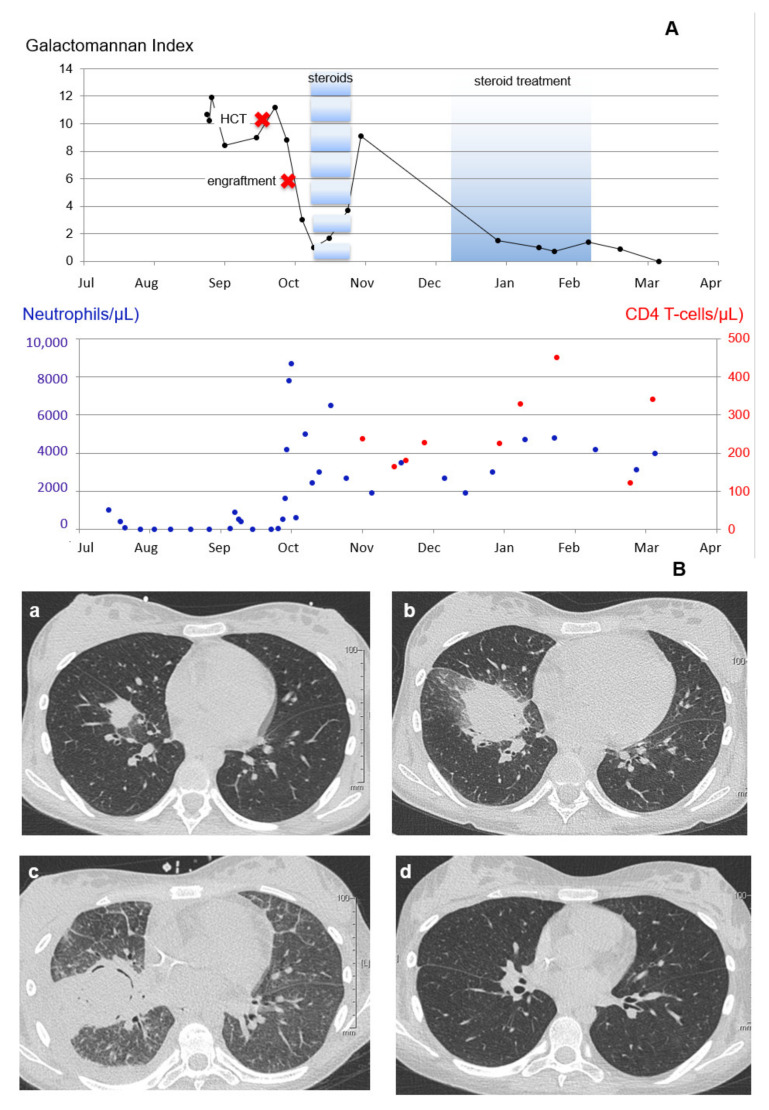
Kinetics of the GM serum antigen index over time in patient 1. (**A**) GM kinetics over time, key clinical events, and corresponding neutrophil and CD4+ T-lymphocyte counts. (**B**) Corresponding pulmonary CT imaging findings in the left lung: (**a**) 8 August; (**b**) 2 September; (**c**) 10 October; and (**d**) 21 March.

**Figure 2 jof-09-00157-f002:**
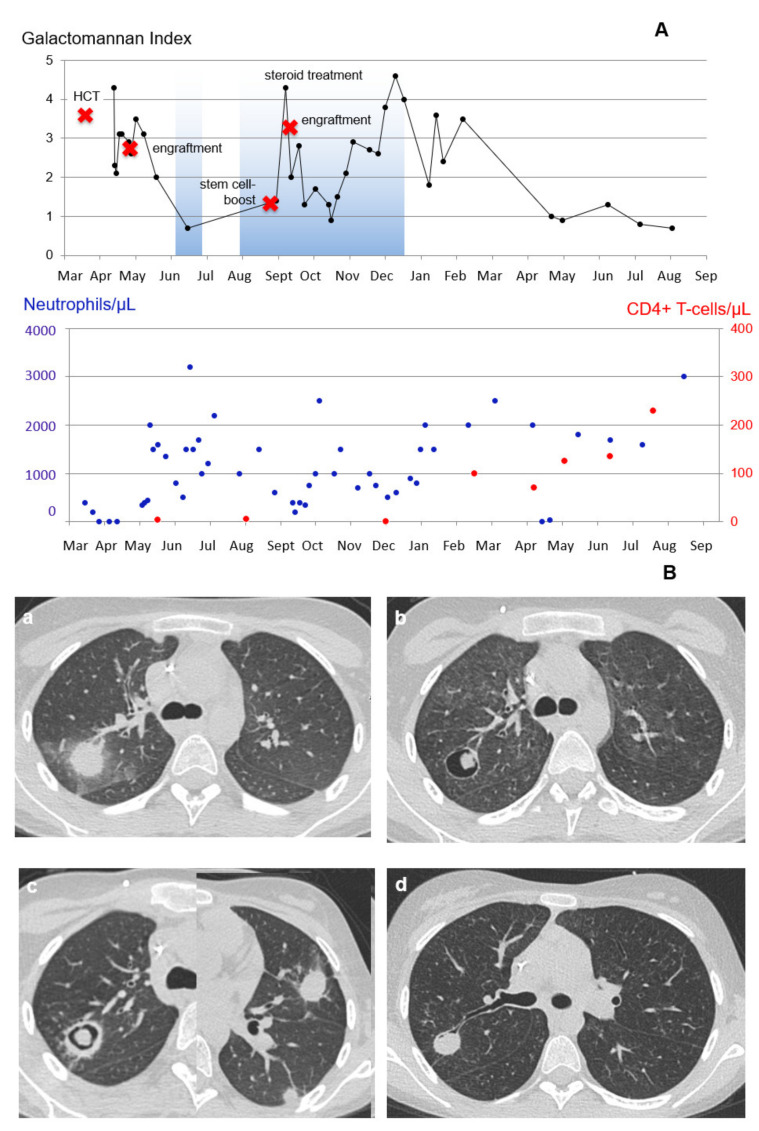
Kinetics of the GM serum antigen index over time in patient 2. (**A**) GM kinetics over time, key clinical events, and corresponding neutrophil and CD4+ T-lymphocyte counts. (**B**) Corresponding CT imaging findings in the left lung: (**a**) April 23; (**b**) June 14; (**c**) August 22 (section with the new lesions in the right lung copied on top of the section of the left lung); and (**d**) June 19.

## Data Availability

The datasets generated during and/or analysed during the current study are available from the corresponding author on reasonable request.

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
