# Peer review of "Long-Term Kinetics of Serum Galactomannan during Treatment of Complicated Invasive Pulmonary Aspergillosis"

_jof, 2023, doi:10.3390/jof9020157_

Round 1

Reviewer 1 Report

This is a good case report in which the correlation between long-term GM follow-up and clinical, radiological and hematological findings is evaluated in pediatric patients. However;

Page 1, line 30-32; You should specify that the amount of galactomannan release by other fungi is considerably lower in comparison to that seen with Aspergillus and Penicillium; otherwise, it may be understood that galactomannan is a pan-fungal marker like beta glucan.

Page 2, line 70; to which criteria did you decide to “probable IPA” in case 1, please detail.

Page 6, line 209-218; it is very long sentence and not understandable; please reorganize.

Please write the abbreviations clearly where it first appears throughout the paper.

There are a few parts needed correction marked yellow on the original paper.

Author Response

Reviewer 1:

Comments and Suggestions for Authors

This is a good case report in which the correlation between  long-term GM follow-up and clinical, radiological and hematological findings is evaluated in pediatric patients. However;
1. Page 1, line 30-32; You should specify that the amount of  galactomannan release by other fungi is considerably lower in comparison to that seen with Aspergillus and Penicillium; otherwise,  it may be understood that galactomannan is a pan-fungal marker like beta glucan.

Response: We agree with the reviewer and have added “in lower amounts” to the listing of non-Aspergillus, and non-Penicillium fungal organisms in the text.

  1. Page 2, line 70; to which criteria did you decide to “probable IPA” in case 1, please detail. (clinical and radiographic criteria plus GM in serum)

Response: As suggested, we added the information: “…based on clinical, mycological (GM) and radiological (CT-scan) criteria..” to explain the diagnosis of probable IPA.

  1. Page 6, line 209-218; it is very long sentence and not understandable; please reorganize.

Response: As recommended, the sentence has been reorganized in two sentences.

  1. Please write the abbreviations clearly where it first appears  throughout the paper.

Response: The text has been reviewed and abbreviation have been added appropriately as recommended.

  1. There are a few parts needed correction marked yellow on the original paper. Response: As we could not find marked text parts, we were unable to make the suggested corrections.

Reviewer 2 Report

Long-term kinetics of serum galactomannan is interesting theme, especially in immunocompromised adolescents with invasive pulmonary aspergillosis. Although there are only two cases of this type of patient, the correlation of many measured points of galactomannan in relation to multiple immune aberrations and the effectiveness of therapy can be very important and across the literature, this scientific point is less represented.

General comments:

-        The data in the supplementary materials do not provide any additional information, they are the same as the figures in the text. I think, the data of supplementary appendix are not needed.

-        Patient no. 2: I´m interested in the outcome of patient no. 2? Eventually, the lesions disappeared completely, or he was transferred to outpatient care, so he could not be monitored until the lesion was completely healed. 

-        I leave it to the authors consideration: Even if only two cases are presented in the article, the correlation analysis (e.g Pearson´s correlation analysis) could be calculated and expressed with linear relationship or linear association number. The evidently strong correlation between two variables (neutrophil count and galactomannan level) could be reported as a specific result and added to the abstract. Moreover, analysis of the correlation results between GM and neutrophil counts would be a relevant supplementary material appropriately cited in the text. 

Minor: 

Line 31: I´m missing a dot at the end of "spp."

Line 31: Fusarium „spp.“ without cursive

Line 42: Cite the reference in the appropriate form if it relates to the meaning of the text. I think the reference may not be related because it expresses a general guide to diagnostic accuracy studies: „Reporting diagnostic accuracy studies: where are we now? “

Line 42: „A positive GM test…“ You mean double positive test? It could be stated because mycological criteria include it. 

Line 43: EORTC/MSGERC, Please, state full name of european organisation study group and abbreviation place in the parenthesis.

Line 46: allogeneic „HCT“ - Explanation of abbreviation need to be stated when it is first mentioned in the text.

Line 51: double „in“, it is redundant.

Line 51: „IPA“I think, abbreviation means "invasive pulmonary aspergillosis" and was not explained before in the text, only in abstract. State it.

Line 59: „… broad spectrum of antibiotics “. It would be worth mentioning the list, dosage, and duration of ATB therapy (e.g. therapeutic and/or prophylaxis possibilities).

Line 67: If abrreviation means "Granulocyte Colony-Stimulating Factor, please state it. Explanation is missing in the text before.

Line 73: ANC in the parenthesis. Is it Absolute Neutrophil Count? Should be stated.

Line 82: „Granulocyte counts were stable. “ Counts should be specified.

Line 84: „…corticosteroids in changing doses…“Specify dosing and its changes in time period.

Line 110: Culture of BALF samples was not performed or is it just missing a mention in the text? I wonder what the sensitivity of A. fumigatus to antifungals was in vitro.

Author Response

Reviewer 2:

Comments and Suggestions for Authors

Long-term kinetics of serum galactomannan is interesting theme, especially in immunocompromised adolescents with invasive pulmonary aspergillosis. Although there are only two cases of this type of patient, the correlation of many measured points of galactomannan in relation to multiple immune aberrations and the effectiveness of therapy can be very important and across the literature, this scientific point is less represented.

General comments:

  1. The data in the supplementary materials do not provide any additional information, they are the same as the figures in the  text. I think, the data of supplementary appendix are not needed.

Response: We agree with the reviewer – all supplementary materials have been removed as suggested. .

  1. Patient no. 2: I´m interested in the outcome of patient no.  2? Eventually, the lesions disappeared completely, or he was  transferred to outpatient care, so he could not be monitored until the lesion was completely healed. 

Response: The lesions eventually  disappeared and patient 2 (as well as patient 1) is in continuous  remission and doing well at +3 years post transplant. We have added this information to the text.

  1. I leave it to the authors consideration: Even if only two  cases are presented in the article, the correlation analysis (e.g  Pearson´s correlation analysis) could be calculated and expressed  with linear relationship or linear association number. The evidently 
    strong correlation between two variables (neutrophil count and  galactomannan level) could be reported as a specific result and  added to the abstract. Moreover, analysis of the correlation results  between GM and neutrophil counts would be a relevant supplementary material appropriately cited in the text.

Response: We understand the reviewers intent to better characterize the relationship between neutrophil count and the galactomannan concentrations in blood. However, given that these are two case reports, we believe that this will not add to the validity of this descriptive observation –no changes.

Minor:

Line 31: I´m missing a dot at the end of "spp."

Response: Corrected as recommended.

Line 31: Fusarium „spp.“ without cursive

Response: Corrected as recommended.

Line 42: Cite the reference in the appropriate form if it relates to 
the meaning of the text. I think the reference may not be related 
because it expresses a general guide to diagnostic accuracy studies: 
„Reporting diagnostic accuracy studies: where are we now? “

Response: We agree with the reviewer. The reference by Leeflang (2015) has been deleted as suggested by the reviewer.

Line 42: „A positive GM test…“ You mean double positive test? It could be stated because mycological criteria include it.

Response: Please note that according to the revised criteria published by Donnelly et al in 2008, a single serum or plasma GM test ≥ 1.0 is a mycological criterion for the diagnosis of probable IA.

Line 43: EORTC/MSGERC, Please, state full name of european 
organisation study group and abbreviation place in the parenthesis.

Response: As suggested, the full names have been added and abbreviations are placed in parenthesis.

Line 46: allogeneic „HCT“ - Explanation of abbreviation need to be 
stated when it is first mentioned in the text.

Response: The full name has been added and the abbreviation is placed in parenthesis as suggested by the reviewer.

Line 51: double „in“, it is redundant.

Response: “in” has been deleted.

Line 51: „IPA“ I think, abbreviation means "invasive pulmonary 
aspergillosis" and was not explained before in the text, only in 
abstract. State it.

Response: We have added the full name and the abbreviation in parenthesis as suggested.

Line 59: „… broad spectrum of antibiotics “. It would be worth 
mentioning the list, dosage, and duration of ATB therapy (e.g. 
therapeutic and/or prophylaxis possibilities). – piperacillin 
tazobactam, the meropenem plus teicoplanin in standard, weight 
adapted doses.

Response: The use of antibiotics as prophylactic, empiric or targeted therapy has been used according to international guidelines. In our opinion, details would not add relevant information to the reader and it would extent the description of the case with information that is not relevant to content and aim of the manuscript – no changes.

Line 67: If abrreviation means "Granulocyte Colony-Stimulating 
Factor, please state it. Explanation is missing in the text before.

Response: As suggested, we have added the full name with the abbreviation in parenthesis.  

Line 73: ANC in the parenthesis. Is it Absolute Neutrophil Count? 
Should be stated.

Response: As suggested, we have added the full name with the abbreviation in parenthesis.

Line 82: „Granulocyte counts were stable. “ Counts should be specified.

Response: As to be seen in the figure the granulocyte counts varied between 2000-4000/μL. This information has been added in the text.

Line 84: „…corticosteroids in changing doses…“Specify dosing and its 
changes in time period.

Response: We agree with the reviewer and have added the precise dosing information.

Line 110: Culture of BALF samples was not performed or is it just 
missing a mention in the text? I wonder what the sensitivity of A. 
fumigatus to antifungals was in vitro.

Response: The cultures from BAL were negative for fungal pathogens, so no AFS testing was feasible. A. fumigatus was detected by PCR, but no molecular testing for resistance was done as this is not part of the routine work-up.